# Identification of Chemical Vapor Mixture Assisted by Artificially Extended Database for Environmental Monitoring

**DOI:** 10.3390/s22031169

**Published:** 2022-02-03

**Authors:** Hi Gyu Moon, Youngmo Jung, Beomju Shin, Donggeun Lee, Kayoung Kim, Deok Ha Woo, Seok Lee, Sooyeon Kim, Chong-Yun Kang, Taikjin Lee, Chulki Kim

**Affiliations:** 1Center for Ecological Risk Assessment, Korea Institute of Toxicology (KIT), Jinju 52834, Korea; higyu.moon@kitox.re.kr (H.G.M.); sykim@kitox.re.kr (S.K.); 2Sensor System Research Center, Korea Institute of Science and Technology (KIST), Seoul 02792, Korea; bilunain@naver.com (Y.J.); bjshin@kist.re.kr (B.S.); donggeun.lee@kist.re.kr (D.L.); kayoungkim@kist.re.kr (K.K.); dockha@kist.re.kr (D.H.W.); slee@kist.re.kr (S.L.); 3Center for Electronic Materials, Korea Institute of Science and Technology (KIST), Seoul 02792, Korea; cykang@kist.re.kr; 4KU-KIST Graduate School of Converging Science and Technology, Korea University, Seoul 02841, Korea

**Keywords:** chemiresistive sensor array, identification of gas mixture, machine learning, support vector machine (SVM), principal component analysis (PCA)

## Abstract

A fully integrated sensor array assisted by pattern recognition algorithm has been a primary candidate for the assessment of complex vapor mixtures based on their chemical fingerprints. Diverse prototypes of electronic nose systems consisting of a multisensory device and a post processing engine have been developed. However, their precision and validity in recognizing chemical vapors are often limited by the collected database and applied classifiers. Here, we present a novel way of preparing the database and distinguishing chemical vapor mixtures with small data acquisition for chemical vapors and their mixtures of interest. The database for individual vapor analytes is expanded and the one for their mixtures is prepared in the first-order approximation. Recognition of individual target vapors of NO_2_, HCHO, and NH_3_ and their mixtures was evaluated by applying the support vector machine (SVM) classifier in different conditions of temperature and humidity. The suggested method demonstrated the recognition accuracy of 95.24%. The suggested method can pave a way to analyze gas mixtures in a variety of industrial and safety applications.

## 1. Introduction

Rapid and reliable detection of chemical vapors is in high demand for applications including environmental monitoring, industrial safety, and national security [1,2,3,4,5,6,7,8]. Since air pollutants are known to cause serious issues in public health and the environment, many efforts in monitoring and analyzing air pollution have been made [9,10]. Accordingly, research has been conducted on the development of various types of sensors, such as a novel X-ray radiation sensor, opto-electronic noses, and electronic noses based on semiconducting metal oxides to detect various gases in real environments [11,12,13,14,15,16]. Among them, an electronic nose equipped with different chemical sensor elements has become the most promising candidate due to high sensitivity, possible integration with high density, and excellent compatibility with conventional electronics [17,18,19,20,21,22,23,24,25]. Despite ongoing research, discrimination of chemical vapors in mixture remains a chronic challenge in the environmental field. The usual difficulties originate from the very nature of various sensing mechanisms in different materials. However, semiconductor-based sensors are not selective to certain target analytes [26,27]. Any sensor material may participate in reactions with a variety of different gas species, allowing for unavoidable cross-sensitivity. Some reactions are also irreversible and poison the sensor, causing it to degrade in terms of sensitivity and reliability. The merit of its high sensitivity is also compromised by the fact that the sensor reactions strongly depend on the environmental conditions such as temperature and relative humidity [28,29,30], rendering measurement reproducibility and analysis consistency inferior.

On the other hand, computing ability has become powerful enough to enable the promise of artificial intelligence. Recent rise of interest in artificial intelligence led to processing responses of various sensor devices to improve their fundamental deficiency in selectivity [31,32,33,34]. Machine-learning algorithms based on support vector machine (SVM) and artificial neural network (ANN) have been applied for making decisions or predictions in a wide variety of applications [35,36,37,38]. Omatu et al. reported on an electronic nose system using a neural network algorithm to identify nine different flavors of coffee and tea [39]. Mohareb et al. reported on ensemble-based SVM classifiers using an electronic nose for quality assessment of beef fillets [40]. However, operation with a typical machine learning algorithm requires an enormous amount of data to train the algorithm and achieve high classification accuracy. In other words, different analytes and their mixtures in different combinations and proportions should be tested a priori in all the possible environmental conditions in terms of temperature and relative humidity [41,42,43]. This becomes a major hurdle to realize the original idea and makes practical application almost impossible. Here, we suggest a novel method to distinguish individual chemical vapors and even their mixtures and demonstrate it with a fully integrated chemiresistive sensor array (ICSA). The suggested protocol is composed of principal component analysis (PCA), artificial database construction, and classification (SVM). Recognition of individual target vapors of NO_2_, HCHO, and NH_3_ and their mixtures is evaluated by applying the SVM classifier in different conditions of temperature and humidity. The suggested method demonstrates a recognition accuracy of 95.24%. Furthermore, this method significantly reduces the required amount of training data by using an artificially extended database for the identification of chemical vapor mixtures with high accuracy.

## 2. Experimental

### 2.1. Chemiresistive Sensor Array

Development of an integrated chemiresistive sensor array (ICSA) has been extensively exploited towards high performance gas sensors [44,45]. Along this line of thought, an ICSA with 16 sensor elements was fabricated out of four metal-oxide films (In_2_O_3_, SnO_2_, WO_3_, and TiO_2_) combined with three catalysts of Au, Pd, and Pt [46], as seen in Figure 1a. To minimize the variability in device performance, sequential semiconductor fabrication processes were applied. The Pt-interdigitated electrode (IDEs) patterns were fabricated using photolithography and dry etching. In detail, organic or inorganic contaminations on a Pt (200 nm)/SiO_2_/Si wafer are usually removed by wet chemical treatment. The photoresist-coated Pt wafer is then prebaked to drive off excess photoresist solvent, typically at 100 °C for 60 s on a hotplate. After prebaking, the photoresist is exposed to a pattern of intense light for 25 s. Positive photoresist of the part exposed to light becomes soluble in the developer. In etching, a plasma (dry) chemical agent removes the uppermost Pt layer of the substrate in the areas that are not protected by the photoresist. The gap distance between electrodes in the fabricated Pt IDEs was 5 μm. The width and thickness of the Pt layer were 40 μm and 200 nm (Figure 1a). The metal oxide thin films (100-nm thick) of WO_3_, SnO_2_, TiO_2_, and In_2_O_3_ were deposited onto predefined regions (1 mm × 1 mm) by using sequential processes of photolithography and in situ electron-beam evaporation. After the deposition of 3-nm-thick functionalizing layers with novel metals (Au, Pt, and Pd), the chemiresistive sensor array (CSA) with 16 sensor elements was realized. Base pressure and applied power for deposition were 2 × 10^−6^ mTorr and 50–70 kW, respectively. The deposition rate was 1.5–3.3 Å/s. After annealing at 500 °C for 2 h, the WO_3_, SnO_2_, TiO_2_, and In_2_O_3_ films were crystallized and a 3 nm-thick metal film was agglomerated into nanoparticles (NPs), resulting in thin films decorated with NPs on the surface. After annealing, the diameter of self-agglomerated metal NPs on the surfaces of In_2_O_3_ and SnO_2_ films were about 10 nm, while it was about 50 nm on TiO_2_ and WO_3_. Such variation in size is attributed to the surface energy difference between metal NPs and metal oxides, where smaller NPs have higher surface energies. These thin films were characterized by X-ray diffraction (XRD), revealing that the as-deposited amorphous films were crystallized to cassiterite crystal phase with tetragonal rutile structure of SnO_2_, a polycrystalline anatase phase of TiO_2_, monoclinic phase of WO_3_, and cubic bixbyite structure of In_2_O_3_. Furthermore, ICSAs were operating at different temperatures of 150 and 200 °C. With this, we obtained 32 different sensing elements because of the temperature dependence of the chemical vapor sensing as well as the enhanced selectivity with noble metals [46,47,48]. The collection of sensor responses lead to a response pattern for a specific analyte, and these patterns provide fingerprints for different gas mixtures (Appendix A). A micro-heater unit was placed under the substrate to thermally activate the sensors. The CSA module has plug-and-play capability in which all the sensor elements in the CSA module underwent an aging process (150 °C or 200 °C) for 72 h to make the sensing materials thermally stable (Figure 1b).

### 2.2. Experiment Conditions

The fabricated ICSA was examined in dry air (5% or less humidity) using a chamber with the volume of 12,800 cm^3^ (16 cm (W) × 16 cm (H) × 50 cm (L)). The measurement system has an injection path where the analyte gas flow is controlled by an auto-functionalized mass flow controller (MFC) and monitored by digital humidity and temperature sensors (SMART SENSOR, AR837). With vapor flow changes from ambient air to target chemical vapors and their mixtures at different concentrations, resistance variations of sensor elements in the ICSA were recorded. The vapor flow rate was controlled in high precision by an automatic mass flow control (MFC) with a mixing chamber installed. The total flow rate for base air and target vapors was 2000 sccm (cm^3^/min) [46]. A micro-heater was mounted on the back of a printed circuit board (PCB) of ICSA, and the power consumption was 250 mW. The operation power of our CSA was about 0.87 mW/mm^2^ (=250 mW/(13 mm × 22 mm)) at 200 °C and 0.77 mW/mm^2^ (=220 mW/(13 mm × 22 mm)) at 150 °C.

### 2.3. Artificial Database

We suggest a data processing protocol and demonstrate it by using the responses obtained from the ICSA. The flow chart of this protocol is shown in Figure 2. Response patterns for target analytes of NO_2_, NH_3_, and HCHO vapors are prepared in the form of a matrix. The response of the ICSA (V_R_) is normalized by the following equation:(1)$$V_{R}=\frac{V_{M}-V_{0}}{V_{0}}\times 100$$
where V_M_ is the maximum voltage and V_0_ is the baseline voltage [49]. The maximum response (V_normal_) is normalized as
(2)$$V_{\mathrm{normal}}=\frac{\max V_{R}-\min V_{R}}{\min V_{R}}$$

The normalized response and the increase or decrease in resistance provide the amplitude and polarity of the input matrices for PCA. The database was then expanded assuming Gaussian distribution of the resistance in the sensing layer. One hundred data samples were randomly generated for a single analyte vapor at a certain concentration in the range of −2σ to +2σ (σ: standard deviation). The time to reach 90% variation in resistance upon exposure to an analyte vapor is defined as the 90% response time. We indicated the maximum response of sensor at a point of 90% response time. In addition, the matrix was not automatically scaled. We performed a normalization step to scale the signal of ICSA. The Min-Max Normalization method was used in this study (Equation (2)). Three individual gases and four mixture gases were identified using 32 sensors. To achieve this, the normalization process was done on the dataset, including all the responses at different concentrations in different combinations of analyte vapors.

In this study, the linear SVM was applied to identify individual and mixture vapors. This classifier represented high identification accuracy for classification of vapors. However, our suggested method is not suitable for applying to nonlinear responses. This is because applied assumptions do not consider reactions between different chemical vapors and take linear translations of the target responses in the principal component space. We also used the linear SVM to identify individual and mixture vapors, which do not require any kind of kernels. We thought the linear SVM was more robust than other classifiers, and the obtained identification accuracy was high (~95.24%) enough for the operation. The artificial database for gas mixtures was constructed by using the expanded database for individual target vapors. Figure 3 shows schematics of the artificial database construction with the expanded database for NO_2_, HCHO, and NH_3_ vapors. Finally, the artificial database for four different vapor mixtures (NO_2_ + NH_3_, NH_3_ + HCHO, HCHO + NO_2_, and NO_2_ + NH_3_ + HCHO) was generated by using the linear combination of the matrices for individual target vapors. For instance, the database for the mixture of NO_2_ and NH_3_ vapors was generated by combining the normalized maximum responses of NO_2_ and NH_3_ vapors according to the following relation:(3)$$V_{NO_{2}/NH_{3}}=K_{1}\;\cdot \;V_{\mathrm{normal}\;NO_{2}}\;+\;K_{2}\;\cdot \;V_{\mathrm{normal}\;NH_{3}}$$
where K_1_ and K_2_ are weighting factors [50]. To determine the weighting factors, the following assumptions were made: (1) the individual vapors in mixture react independently, and (2) the response matrices for the vapor mixtures can be described as the linear combination of the ones for individual analytes in the first-order approximation. With these, the weighting factors were determined in such a way that the linear combination of the medians of the expanded database for individual vapor responses was translated to the obtained response to the vapor mixtures. The obtained analytical matrix had a 300 by 32 structure for each vapor. The artificial database was used as the training dataset for the SVM classifier.

### 2.4. Measurements

The analyte concentration was varied by controlling the ratio between the analyte gas and dry air. All the measurements in the identical experimental condition were repeated three times. Typical response curves of the ICSA under exposure to NO_2_, NH_3_, and their mixtures at 150 °C are shown in Figure 4a–c. The applied concentrations of NO_2_ and NH_3_ vapors were 2, 5, and 10 ppm. The ones in their mixtures were 2/10, 5/5, and 10/2 ppm (NO_2_/NH_3_). The response curves of other individual vapors and their mixtures at different temperatures are given in Appendix A. As seen in Figure 4a,b, the responses in terms of amplitude and polarity are significantly different according to target vapors.

### 2.5. PCA

The PCA results using the obtained responses to individual vapors and their mixtures are shown in Figure 5a,b, where the ones for vapor mixtures (NO_2_ + NH_3_, NH_3_ + HCHO, HCHO + NO_2_, and NO_2_ + NH_3_ + HCHO) were projected onto the space defined by those principal components for individual vapors (Figure 5b). The features were extracted after the PCA. The percentages of variance by PC1 and PC2 were 92.3% and 0.8%, respectively. We note that the three individual vapors and vapor mixtures were well clustered on the PCA plot, as shown in Appendix A. The collected feature for PCA was the amplitudes in the response to analytes. The maximum responses of 32 sensing elements to 9 different target analytes were obtained. The PCA result using the obtained responses to individual vapors is shown in Figure 5a. We determined the number of main components by considering the percentage of variance and identification accuracy. The data matrix has 32 columns in structure, and accordingly, 32 PCs were obtained in PCA. The percentage of variance indicating the loss of information in converting the data into the principal component space was determined by the calculation of eigenvalues. From the calculation, it turned out that PC1 and PC2 determines 93.1% of the response pattern. Therefore, we decided to use PC1 and PC2 for the suggested protocol. When we confirmed the identification accuracy using higher order PCs, the identification accuracy was not improved. The matrix for each analyte vapor at a certain concentration has 100 by 32 elements in structure after the data expansion process. Since we worked with the data at three different concentrations and tried to differentiate 7 cases (3 individual analyte vapors and 4 different combinations of them), this makes it a 2100 by 32 matrix structure. All the results for mixtures were projected onto the same principal component space [18,46].

## 3. Results and Discussion

### 3.1. Response of Vapor Mixtures

To evaluate the sensing capability of the ICSA, it was placed in a designed chamber with the volume of 12,800 cm^3^. The relative humidity and temperature conditions were consistently monitored through the entire measurements using a thermohydrometer (Dwyer, Model RP2 Thermo-Hygrometer Probe). The total vapor flow was kept at 2000 sccm cm^3^/min by using a programmable MFC. The sensing capability of the ICSA was evaluated under exposure to different chemical vapors, including 10 ppm NH_3_, 10 ppm NO_2_, and NH_3_ + NO_2_ (1:1). The obtained responses were normalized as R = (V_gas_ − V_0_)/V_0_ × 100 (%) for oxidizing gases and R = (V_0_ − V_gas_)/V_0_ × 100 (%) for reducing gases, where V_0_ and V_gas_ denote the initial voltage of the sensor in air and the obtained voltage in response to the analyte. The ICSA exhibited outstanding sensitivity to NO_2_ vapor. The theoretical detection limit for NO_2_ and NH_3_ + NO_2_ (except for Ch 1 and 2) were evaluated to be in the range of 609–896 ppt and 30.4–109 ppb via linear extrapolation. The result is that the response decreases due to mutual interference between oxidizing and reducing vapor and is more dependent on NO_2_ vapor. This was attributed to the dominant presence of an oxidizing agent (NO_2_^−^) at a relatively low temperature [46].

### 3.2. Identification of Vapor Mixtures

In this study, expanding the dataset was required to obtain sufficient resources to enhance the accuracy of identification of analytes in the mixture. Each class had only nine data points and we constructed three test datasets, three training datasets, and three validation datasets by an extended dataset method. The expanded matrices for NO_2_, NH_3_, and HCHO are plotted on the PCA plane shown in Figure 6, where the expanded database (marked in hollow) is overlaid. The expanded database for the individual vapors is well matched to the experimentally obtained values on the PCA plot. The artificial database for vapor mixtures is constructed by the database expansion and their combinations assuming Gaussian distribution of sensor responses and chemical reactions in the linear regime as the first-order approximation. The constructed artificial database of vapor mixtures is marked on the same principal component (PC) plane as shown in Figure 7.

Recognition of individual target vapors of NO_2_, HCHO, and NH_3_ and their mixtures was evaluated by applying the SVM classifier which was trained by the previously obtained artificial database [51]. The total accuracy was 95.24%. As demonstrated, the vapor mixture of NO_2_, HCHO, and NH_3_ was recognized with good accuracy, although the response patterns obtained in ICSA were quite similar (Appendix A). The number of test sets and their identification accuracy under the suggested protocol is charted in the Appendix A. Individual vapors were identified with 100% accuracy, but mixture vapors were 91.67%. Identification for mixture vapors involving HCHO vapor was lower accuracy than others. 

This high level of recognition ability is attributed to the artificially extended database and the applied classifier for the given task. That is, the artificial database apparently covers the cases which were not experimentally tested out, and the hyperplane defined by the SVM improves the accuracy of the identification [51]. The suggested protocol was also shown to be robust against the drift of the response baseline. Although the applied assumptions were a bit primitive in the sense that they ignored the nonlinear nature of chemical vapor reactions in mixture, the suggested protocol demonstrated the recognition of individual vapors and their mixtures with a high level of accuracy. We believe that this can be the first step towards identification of vapor analytes in mixture.

## 4. Conclusions

In summary, we designed a sensor array comprising a fabricated CSA with operating circuitry and demonstrated identification of vapor analytes in mixture. With its high sensing ability, we suggested a machine-learning-assisted recognition of chemical vapor mixtures. The obtained database for individual vapors was expanded by assuming Gaussian distribution of chemical vapor responses. The artificial database was then constructed for vapor mixtures in the first-order approximation. The SVM classifier was employed to evaluate the recognition accuracy for chemical vapors using the artificial database (720 train data, 180 validation data). The suggested protocol recognized target analytes and their mixtures with an accuracy of 95.24%, demonstrating a novel strategy towards ultimate sensing of gas mixtures.

## Figures and Tables

**Figure 1 sensors-22-01169-f001:**
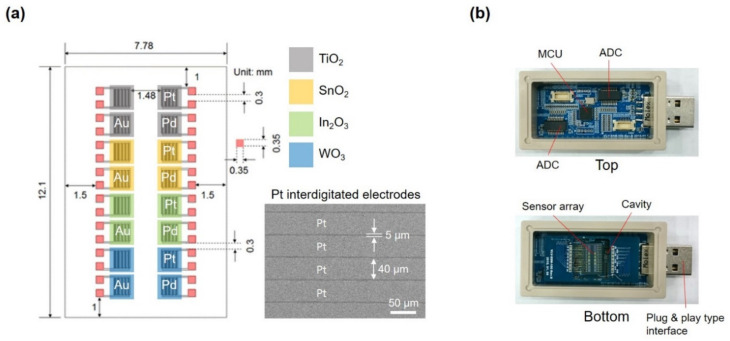
(**a**) Schematic illustration of the configuration of the chemiresistive sensor array (CSA). (**b**) Photographs of the upper and lower circuit boards in the CSA module with plug-and-play capability.

**Figure 2 sensors-22-01169-f002:**
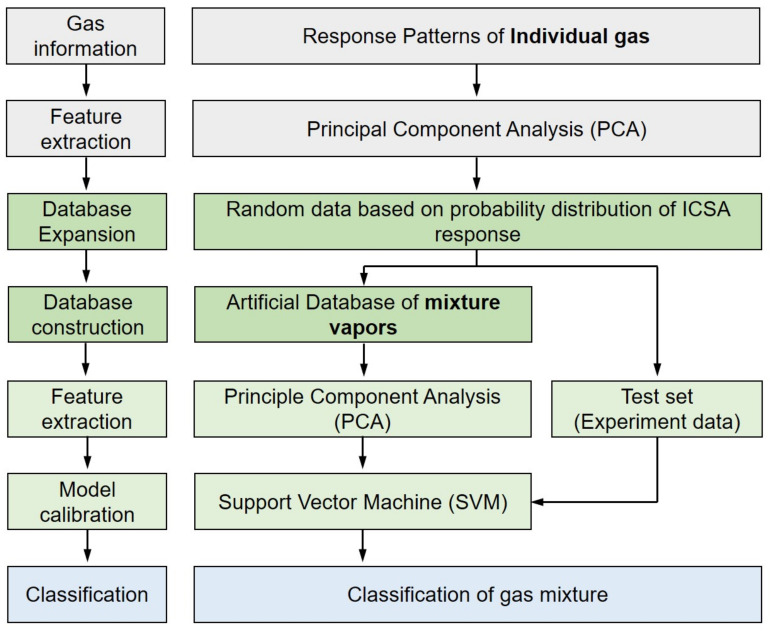
Data flow chart for identification of chemical vapor mixtures.

**Figure 3 sensors-22-01169-f003:**
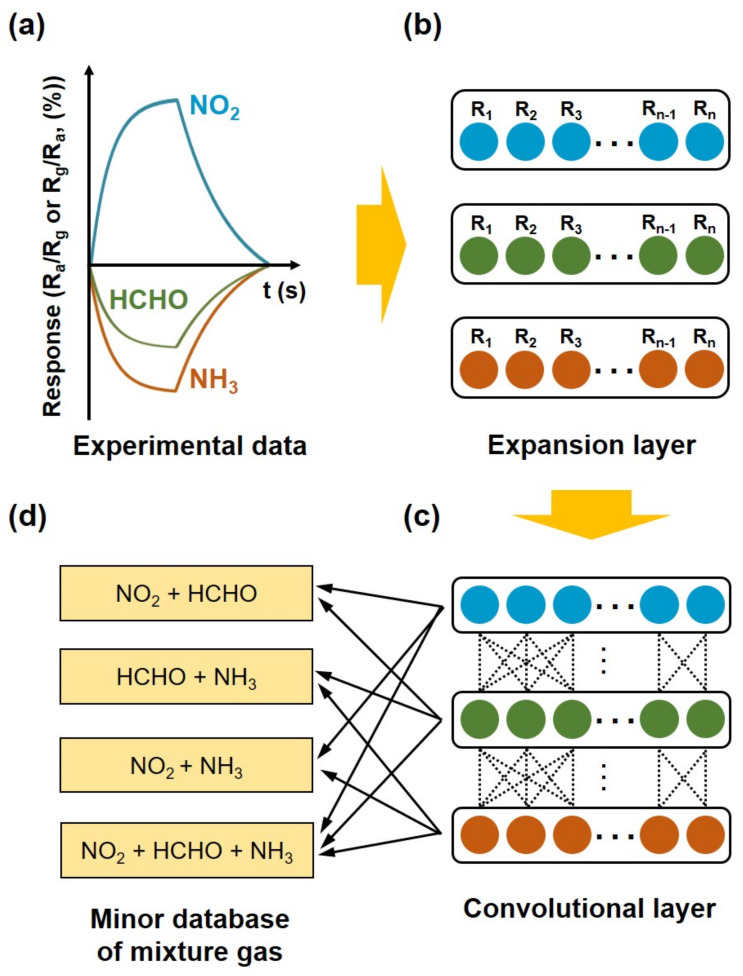
Schematics of artificial database construction for identification of chemical vapor mixtures. (**a**) Measurement of sensor responses to different analyte vapors. (**b**) Expansion layer construction by applying Gaussian distribution. (**c**) Feature definition of mixture gas by a convolution process. (**d**) Minor database construction for mixture gas by a convolutional layer.

**Figure 4 sensors-22-01169-f004:**
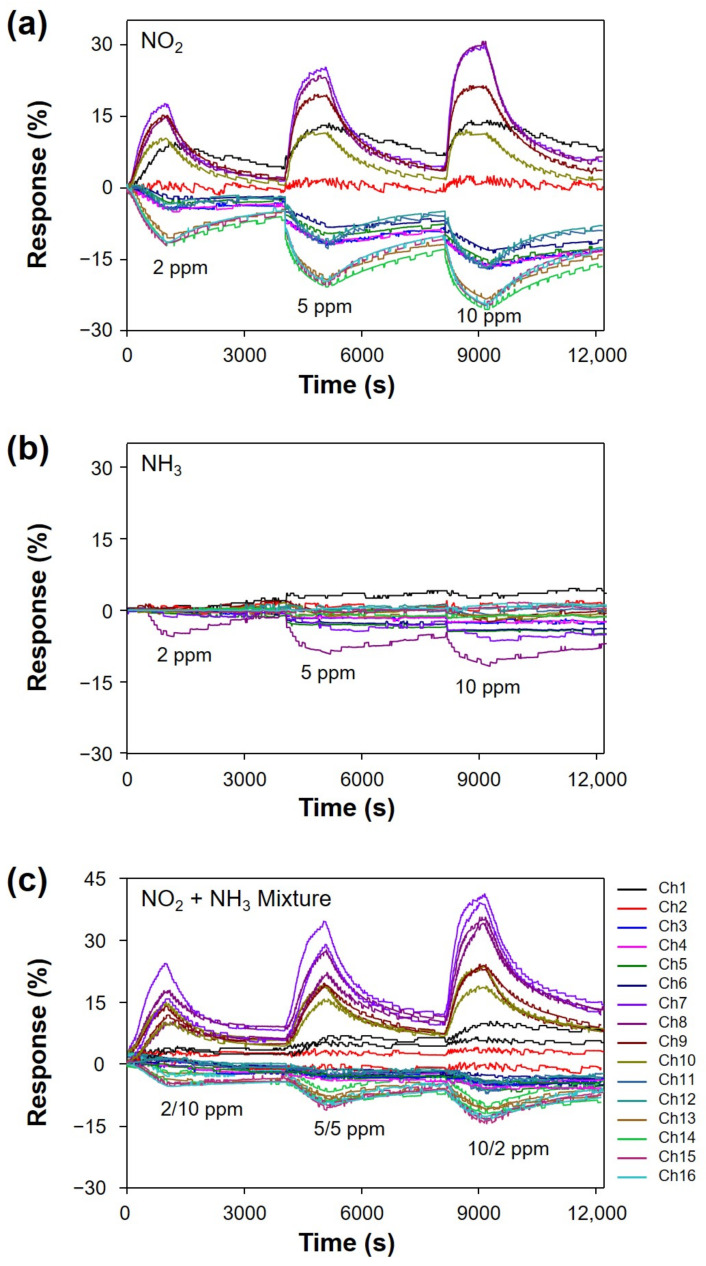
Response curves of the ICSA for (**a**) NO_2_, (**b**) NH_3_, and (**c**) mixtures of NO_2_ and NH_3_ in the concentration range of 2–10 ppm at 150 °C.

**Figure 5 sensors-22-01169-f005:**
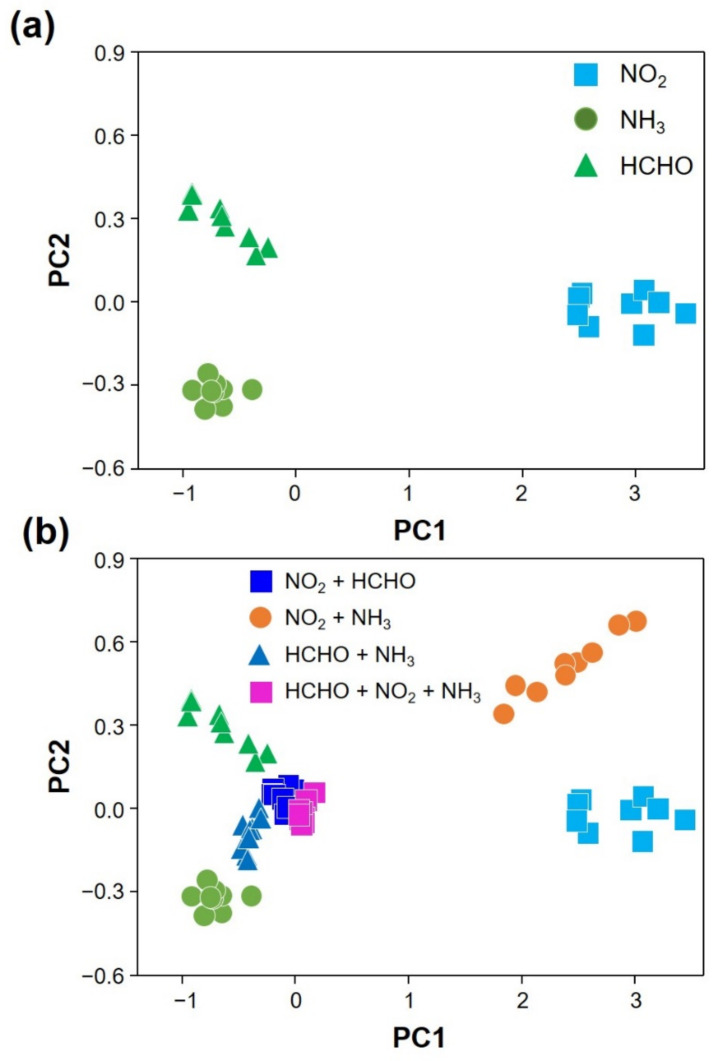
PCA plots for (**a**) individual vapors (NO_2_, NH_3_ and HCHO) and (**b**) their mixtures (NO_2_ + HCHO, NO_2_ + NH_3_, HCHO + NH_3_, and HCHO + NO_2_ + NH_3_).

**Figure 6 sensors-22-01169-f006:**
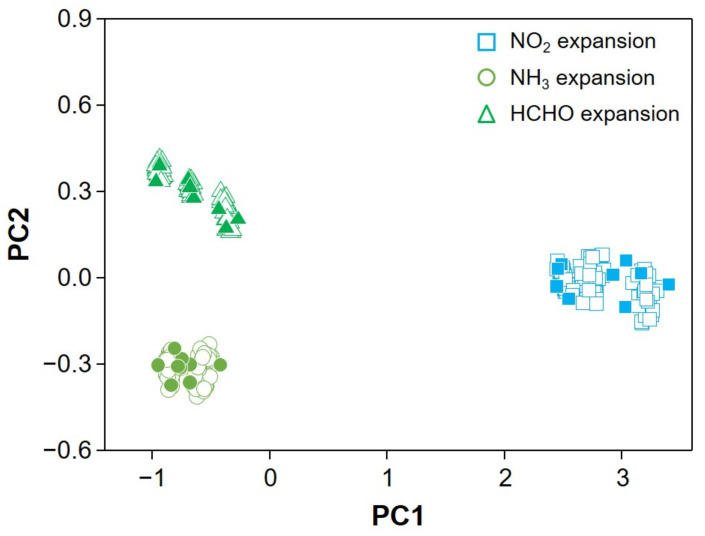
PCA plot of the expanded database for individual vapors.

**Figure 7 sensors-22-01169-f007:**
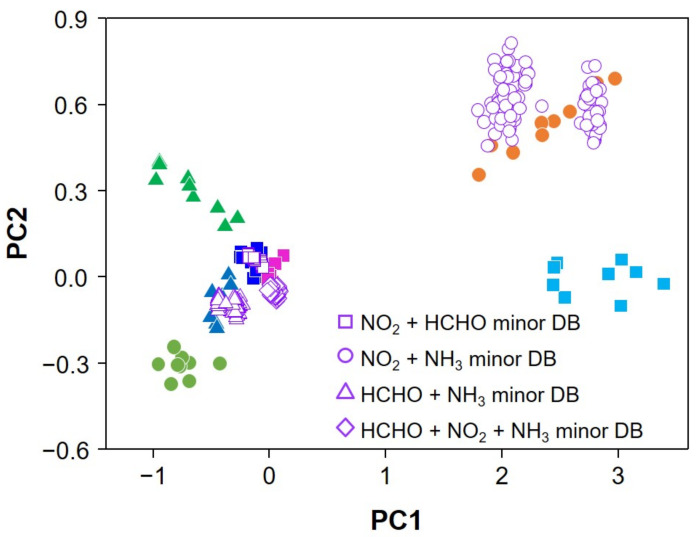
PCA plot of experimentally obtained data and the artificial database for vapor mixtures.

## Data Availability

Not applicable.

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
