# Peer review of "Identification of Chemical Vapor Mixture Assisted by Artificially Extended Database for Environmental Monitoring"

_sensors, 2022, doi:10.3390/s22031169_

Round 1
Reviewer 1 Report
This paper shows that the proposed process is useful as an application of environmental monitoring by machine learning. In results and discussion in Chapter 3, only one sub-item is mentioned, but since the analysis process is multi-item. So, please, add more. In particular, explain that it is difficult to predict non-linear phenomena by citing some references.
Author Response
This paper shows that the proposed process is useful as an application of environmental monitoring by machine learning. In results and discussion in Chapter 3, only one sub-item is mentioned, but since the analysis process is multi-item. So, please, add more. In particular, explain that it is difficult to predict non-linear phenomena by citing some references.
We thank the reviewer for the comment. In this study, the linear support vector machine (SVM) was applied to identify individual and mixture vapors. This classifier represented high identification accuracy for classification of vapors. However, our suggested model is not suitable for applying to non-linear phenomena. It is because we made linear assumptions that we did not take into account reactions between different chemical vapors and the model considers only the linear translation of the target responses in the principal component space. And these assumptions would be valid in the case, for instance, that the concentrations of the analytes are very low. To clarify this, we revised the manuscript.
(Page 4, line 146)
In this study, the linear SVM was applied to identify individual and mixture vapors. This classifier represented high identification accuracy for classification of vapors. However, our suggested model is not suitable for applying to non-linear phenomena. It is because we made linear assumptions that we did not take into account reactions between different chemical vapors and the model considers only the linear translation of the target responses in the principal component space.

Reviewer 2 Report
The authors present an identification of chemical vapor mixture assisted by artificially extended database for environmental monitoring. The research idea is very well conceived and the manuscript is well structured and written. I believe the manuscript would be a good contribution to Sensors. Hence, I would like to accept the manuscript for publication after minor revisions of the following comments.
1- Explain some of the most important results in abstract and last paragraph of introduction section.
2- Briefly explain the fabrication process/technique was used for the deposition of metal oxide thin films?
3- For readers' understanding, provide the dimensions (finger length and gap between adjoining fingers) of the interdigitated electrodes in the experimental section.
4- Provide the equation umbers in a chronological order.
5- I recommend explaining the measurement section of the manuscript into the results and discussion.
Author Response
The authors present an identification of chemical vapor mixture assisted by artificially extended database for environmental monitoring. The research idea is very well conceived and the manuscript is well structured and written. I believe the manuscript would be a good contribution to Sensors. Hence, I would like to accept the manuscript for publication after minor revisions of the following comments.
- Explain some of the most important results in abstract and last paragraph of introduction section.
We appreciate the reviewer’s comment. We agree that the achievement of this work needs to be mentioned as suggested. The database for individual vapor analytes is expanded and the one for their mixtures is prepared in the first order approximation. Recognition of individual target vapors of NO2, HCHO, and NH3 and their mixtures was evaluated by applying the support vector machine (SVM) classifier in different conditions of temperature and humidity. The suggested method demonstrated the recognition accuracy of 95.24%. As suggested, we added the corresponding sentence in the revised manuscript.
(Page 1, line 22 in Abstract)
The database for individual vapor analytes is expanded and the one for their mixtures is prepared in the first order approximation. Recognition of individual target vapors of NO2, HCHO, and NH3 and their mixtures was evaluated by applying the support vector machine (SVM) classifier in different conditions of temperature and humidity. The suggested method demonstrated the recognition accuracy of 95.24%.
(Page 2, line 66 in Introduction)
Recognition of individual target vapors of NO2, HCHO, and NH3 and their mixtures was evaluated by applying the SVM classifier in different conditions of temperature and humidity. The suggested method demonstrated the recognition accuracy of 95.24%.
- Briefly explain the fabrication process/technique was used for the deposition of metal oxide thin films?
We thank the reviewer for the comment. The SiO2/Si substrates were patterned to define Pt interdigitated electrodes in a 4-inch wafer scale by using photolithography and dry etching (Oxford Instrument, Reactive Ion Etching (RIE)). The metal oxide thin films (100-nm thick) of WO3, SnO2, TiO2, and In2O3 were deposited onto predefined regions (1 mm×1mm) by using sequential processes of photolithography and in situ electron-beam evaporation. After the deposition of 3-nm-thick functionalizing layers with novel metals (Au, Pt, and Pd), the chemiresistive sensor array (CSA) with 16 sensor elements was realized. The base pressure and applied power for deposition were 2×10–6 mTorr and 50–70 kW, respectively. The deposition rate was 1.5–3.3 Å/s. The fabricated CSA was annealed at 500 ℃ in air for 120 min to crystallize the amorphous metal-oxide films and an aging process for 72 h was followed. The functionalized layer of the novel metals led to the formation of nanoscale metallic islands after thermal processes. Consequently, we added the corresponding sentence in the revised manuscript as follows.
(Page 2, line 86)
The metal oxide thin films (100-nm thick) of WO3, SnO2, TiO2, and In2O3 were deposited onto predefined regions (1 mm×1mm) by using sequential processes of photolithography and in situ electron-beam evaporation. After the deposition of 3-nm-thick functionalizing layers with novel metals (Au, Pt, and Pd), the chemiresistive sensor array (CSA) with 16 sensor elements was realized. The base pressure and applied power for deposition were 2×10–6 mTorr and 50–70 kW, respectively. The deposition rate was 1.5–3.3 Å/s.
- For readers' understanding, provide the dimensions (finger length and gap between adjoining fingers) of the interdigitated electrodes in the experimental section.
We thank the reviewer for the helpful comment. The gap distance between electrodes in the fabricated Pt IDEs was 5 μm. The width and thickness of the Pt layer were 40 μm and 200 nm. The information on the dimensions of the IDEs and an SEM image were added in the revised manuscript as follows.
(Page 2, line 84)
The gap distance between electrodes in the fabricated Pt IDEs was 5 μm. The width and thickness of the Pt layer were 40 μm and 200 nm.
- Provide the equation umbers in a chronological order.
We thank the reviewer for the comment. We revised the manuscript accordingly.
- I recommend explaining the measurement section of the manuscript into the results and discussion.
We thank the reviewer for the useful comment. To evaluate the sensing capability of the integrated chemiresistive sensor array (ICSA), it was placed in a designed chamber with the volume of 12800 cm3. The relative humidity and temperature conditions were consistently monitored through the entire measurements using a thermohydrometer (Dwyer, Model RP2 Thermo-Hygrometer Probe). The total vapor flow was kept at 2000 sccm cm3/min by using a programmable mass flow controller (MFC). The sensing capability of the ICSA was evaluated under exposure to different chemical vapors including 10 ppm NH3, 10 ppm NO2 and NH3+NO2(1:1). The obtained responses were normalized as R= (Vgas-V0)/V0 ×100 (%) for oxidizing gases and R= (Vo-Vgas)/V0 ×100 (%) for reducing gases, where V0 and Vgas denote the initial voltage of the sensor in air and the obtained voltage in response to the analyte. The ICSA exhibited outstanding sensitivity to NO2 vapor. The theoretical detection limit for NO2 and NH3+NO2 (except for Ch 1 and 2) were evaluated to be in the range of 609–896 ppt and 30.4–109 ppb via linear extrapolation. The result is that the response decreases due to mutual interference between oxidizing and reducing vapor, and is more dependent on NO2 vapor. This attributed to the dominant presence of an oxidizing agent (NO2−) at a relatively low temperature. Consequently, we added the corresponding sentence in the revised manuscript as follows.
(Page 8, line 216)
3.1. Response of vapor mixtures.
To evaluate the sensing capability of the ICSA, it was placed in a designed chamber with the volume of 12800 cm3. The relative humidity and temperature conditions were consistently monitored through the entire measurements using a thermohydrometer (Dwyer, Model RP2 Thermo-Hygrometer Probe). The total vapor flow was kept at 2000 sccm cm3/min by using a programmable mass flow controller (MFC). The sensing capability of the ICSA was evaluated under exposure to different chemical vapors including 10 ppm NH3, 10 ppm NO2 and NH3+NO2(1:1). The obtained responses were normalized as R= (Vgas-V0)/V0 ×100 (%) for oxidizing gases and R= (Vo-Vgas)/V0 ×100 (%) for reducing gases, where V0 and Vgas denote the initial voltage of the sensor in air and the obtained voltage in response to the analyte. The ICSA exhibited outstanding sensitivity to NO2 vapor. The theoretical detection limit for NO2 and NH3+NO2 (except for Ch 1 and 2) were evaluated to be in the range of 609–896 ppt and 30.4–109 ppb via linear extrapolation. The result is that the response decreases due to mutual interference between oxidizing and reducing vapor, and is more dependent on NO2 vapor. This attributed to the dominant presence of an oxidizing agent (NO2−) at a relatively low temperature [46].

Reviewer 3 Report
The manuscript describes an electronic nose made with thin films of 4 different metal oxides decorated with different metal nanoparticles (and operating at 2 different temperatures). The electronic nose is able to correctly classify the 4 single gases and their mixtures. The results presented are interesting, but the presentation is lacking above all in two respects: 1) syntax & grammar and 2) scientificity. A scientific article cannot avoid describing the steps and details of the experimental procedures. Any research group should be able to replicate the experiments, but in this case it is impossible because the text is often lacking or vague.
Here is a list of more specific comments, divided into major and minor. Here are the major ones:
- In addition to mere oversights, there are frequent grammar and syntax errors, including important ones (such as singular/plural coherence and missing verb in the sentence, even in the abstract). Here are a few examples, but I suggest submitting the manuscript to a native speaker or professional service.
- Line 53: “requires an enormous amount of database”. Probably you meant data, not database.
- Line 58: “to distinguish chemical vapors of individuals and even their misture” should probably be “to distinguish individual chemical vapors and even their mixtures”.
- Line 66: “Our 16 sensor elements of ICSA is composed of four…” The sentence structure is almost intelligible.
Oversights:
- Line 54: “alg,korithm” should be “algorithm”
- Line 57: “origianl” should be “original”
- Line 100: “targent” should be “target”
- Line 70-73: you pass from the realization of the Pt electrodes to the final annealing without explaining the steps in between (first the deposition of the MO films and then the surface decoration with metal nanoparticles). The various sections of the "experimental" must be integrated by explaining the procedures in detail: a scientific article should give enough information that any other group can reproduce the experiments.
- Line 99-100: “The maximum responses”: how did you calculate the maximum response? It is usually calculated when the response has reached saturation, but from the graphs it seems that it does not reach it. If the answer is reached after a certain time, it is better to specify this definition.
- Lines 106 - 113: normalization is not well explained: on which set of values is it performed? It seems to me on the single measurement (all sensors for a certain concentration of a certain gas), but it is important to understand it. And why is there no trace of this normalization in Fig. S1?
- 2: in the second line it seems that the features are extracted after the PCA, not from the real response values of the sensors. However, this passage is not described in the text. In addition, there is a second PCA for the data relating to the mixtures. Also, if you use the values after PCA, how many PCs do you have? All 32 or only the most significant?
- Three different ratios (10-2, 5-5 and 2-10) were measured for each mixture, but they were then classified together as a single class. Why is a 10-2 mixture classified together with the 2-10 mixture instead of 10-0 (i.e. pure gas)?
- How are the various datasets (train, validation, test) made for the SVM? On line 172 it is written that the SVM is trained with an artificial database. It is important to understand how many data and which ones (experimental or artificial) each dataset contains.
- Only one quantitative result is given regarding the classification: on line 173 it is said that the total accuracy is 95.24%. The authors should specify the accuracy for each type of measurement: each individual gas and each of the mixtures, to understand which classes are easier and more difficult to discriminate.
- The plots in Fig. S3 show strong drift, higher than the response and recovery of the various measures. Why does this not negatively affect the performance of the electronic nose?
- The SVM classifier is not described in detail (for example: what kind of kernel does it use?)
Minor comments:
- Line 69: CSA is not defined (you have defined ICSA some lines before). It would be better to use homogeneous terminology throughout the article.
- Line 72-73: "forming Au, Pd, and Pt islands in nanoscale via the agglomeration of the Au films." I do not think that the agglomeration of Au can create islands of Pd or Pt.
- Line 77: “ICSAs were operating at different temperatures” : are they two different arrays or just one that operated first at one temperature and then at another temperature?
- Line 84: “were aged”: in what conditions were they aged? At normal working condition (150 or 200 ° C?) Or at higher temperature to stabilize the crystal structure?
- Line 88: what does “auto-functionalized mass flow controller” mean?
- Line 89: "With vapor flow change from ambient air to target chemical vapors": but at the beginning of the paragraph you say that you used dry air. Have you used ambient air or dry air?
- Line 109: “when the response is saturated”: in Fig. 4 it does not seem that saturation is ever reached
- After Fig. 1, Fig. 5 is mentioned in the text. Perhaps it would be worth changing the order of the figures so that they go in order with the text.
- Line 112 "The normalized response and its direction give the amplitude and polarity of the input matrices for PCA." What does “direction” mean? You probably mean if the resistance increases or decreases, but since you made the absolute value in the equation on line 108, the Vnormal has no "direction". And what are the “amplitude and polarity” of matrices for PCA? The matrix for the PCA should simply consist of 9 lines for each gas, each line consisting of the 32 response values obtained from the various sensors.
- Line 114: “One hundred data samples were randomly generated…”: one hundred in total or one hundred for each gas?
- The random error to generate virtual data is applied to each individual sensor individually, and not correlated with the error on the others, right?
- Line 130: "The obtained analytical matrix has 300 by 32 structure for each vapor." -> do you mean for each mixture? What about the 3-gas mixture?
- Fig S4: the labels of the lines are missing (I imagine they are the 32 sensors composing the array, but they should be specified to better understand the effect of material, decoration and working temperature on the response.
- Principal components are ordered with the% variance they bear. The difference between PC1 and PC2 is very high (92.3 and 0.8% respectively). Can the authors explain this behaviour between the importance of PC1 and all 31 other PCs?
- Lines 162 - 164: from the 9 experimental points for each gas you have created 91 "virtual" points for each gas. How are they divided into test, training and validation datasets? The points in Fig. 6 are the test ones, I guess.
Author Response
The manuscript describes an electronic nose made with thin films of 4 different metal oxides decorated with different metal nanoparticles (and operating at 2 different temperatures). The electronic nose is able to correctly classify the 4 single gases and their mixtures. The results presented are interesting, but the presentation is lacking above all in two respects: 1) syntax & grammar and 2) scientificity. A scientific article cannot avoid describing the steps and details of the experimental procedures. Any research group should be able to replicate the experiments, but in this case it is impossible because the text is often lacking or vague.
Here is a list of more specific comments, divided into major and minor. Here are the major ones:
- In addition to mere oversights, there are frequent grammar and syntax errors, including important ones (such as singular/plural coherence and missing verb in the sentence, even in the abstract). Here are a few examples, but I suggest submitting the manuscript to a native speaker or professional service.
Thank you for the comments. In response to this reviewer’s comment, we completed the English correction as a whole with the help of a professional service.
- ine 53: “requires an enormous amount of database”. Probably you meant data, not database.
Thank you for the comments. We corrected the sentence in the revised manuscript.
- Line 58: “to distinguish chemical vapors of individuals and even their misture” should probably be “to distinguish individual chemical vapors and even their mixtures”.
Thank you for the comments. We clarified the corresponding sentence in the revised manuscript.
- Line 66: “Our 16 sensor elements of ICSA is composed of four…” The sentence structure is almost intelligible.
Thank you for the comments. We clarified the corresponding sentence in the revised manuscript.
Oversights:
- Line 54: “alg,korithm” should be “algorithm”
Thank you for the comments. We corrected the typo in the revised manuscript.
- Line 57: “origianl” should be “original”
Thank you for the comments. We corrected the typo in the revised manuscript.
- Line 100: “targent” should be “target”
Thank you for the comments. We corrected the typo in the revised manuscript.
- Line 70-73: you pass from the realization of the Pt electrodes to the final annealing without explaining the steps in between (first the deposition of the MO films and then the surface decoration with metal nanoparticles). The various sections of the "experimental" must be integrated by explaining the procedures in detail: a scientific article should give enough information that any other group can reproduce the experiments.
We thank the reviewer for this helpful comment. The Pt interdigitated electrodes (IDEs) patterns were fabricated using photolithography and dry etching. In detail, organic or inorganic contaminations on Pt (200 nm)/SiO2/Si wafer are usually removed by wet chemical treatment. The photoresist-coated Pt wafer is then prebaked to drive off excess photoresist solvent, typically at 100 °C for 60 s on a hotplate. After prebaking, the photoresist is exposed to a pattern of intense light for 25 s. Positive photoresist of the part exposed to light becomes soluble in the developer. In etching, a plasma (dry) chemical agent removes the uppermost Pt layer of the substrate in the areas that are not protected by photoresist. The fabricated chemiresistive sensor array (CSA) was fabricated using a SiO2/Si substrate with Pt IDEs in which the gap between electrodes was 5 μm. The width and thickness of Pt IDEs were 40 μm and 200 nm. Consequently, we added the corresponding sentence and SEM image in Fig.1 in the revised manuscript as follows.
(Page 2, line 77)
The Pt interdigitated electrodes (IDEs) patterns were fabricated using photolithography and dry etching. In detail, organic or inorganic contaminations on Pt (200 nm)/SiO2/Si wafer are usually removed by wet chemical treatment. The photoresist-coated Pt wafer is then prebaked to drive off excess photoresist solvent, typically at 100 °C for 60 s on a hotplate. After prebaking, the photoresist is exposed to a pattern of intense light for 25 s. Positive photoresist of the part exposed to light becomes soluble in the developer. In etching, a plasma (dry) chemical agent removes the uppermost Pt layer of the substrate in the areas that are not protected by photoresist. The fabricated chemiresistive sensor array (CSA) was fabricated using a SiO2/Si substrate with Pt IDEs in which the gap between electrodes was 5 μm. The width and thickness of Pt IDEs were 40 μm and 200 nm. (Figure 1 (a)).
- Line 99-100: “The maximum responses”: how did you calculate the maximum response? It is usually calculated when the response has reached saturation, but from the graphs it seems that it does not reach it. If the answer is reached after a certain time, it is better to specify this definition.
We appreciate the reviewer for the sharp comment. In general, the times to reach 90% (τ90) variation in resistance upon exposure to detecting gas and air are defined as the 90% response time, as shown in Fig.1 below. We indicated the response of sensor at a point of 90% response time using the following formula: R= (Vgas-V0)/V0 ×100 (%) for oxidizing gases. As mentioned by the reviewer, we added the corresponding sentence in the experimental section of the revised manuscript as follows.
(Page 4, line 139)
The times to reach 90% variation in resistance upon exposure to detecting gas and air are defined as the 90% response time. We indicated the maximum response of sensor at a point of 90% response time.
- Lines 106 - 113: normalization is not well explained: on which set of values is it performed? It seems to me on the single measurement (all sensors for a certain concentration of a certain gas), but it is important to understand it. And why is there no trace of this normalization in Fig. S1?
We thank the reviewer for the useful comment. In the study, normalization is based on the response amplitudes of all the sensors upon exposure to analyte gases. The normalization was performed on all the obtained data at different concentrations in different combinations of analyte vapors by using the Min-Max Normalization method. The normalization equation is described as below:
We agree with the reviewer that the addition of the normalization in Fig. S1 is needed. For this, we added the trace in Fig. S1.
- 2: in the second line it seems that the features are extracted after the PCA, not from the real response values of the sensors. However, this passage is not described in the text. In addition, there is a second PCA for the data relating to the mixtures. Also, if you use the values after PCA, how many PCs do you have? All 32 or only the most significant?
We thank the reviewer for the useful comment. As suggested, we added the sentence saying that the features were extracted after the PCA. In this study, we considered all 32 PCs based on 32 sensors. From the calculation of the variance percentages of PCs, we found out that PC1 and PC2 were the most significant where the information loss rate was only 6.9%. Therefore, we utilized those two PCs in the suggested protocol.
(Page 6, line 190)
The features were extracted after the PCA.
- Three different ratios (10-2, 5-5 and 2-10) were measured for each mixture, but they were then classified together as a single class. Why is a 10-2 mixture classified together with the 2-10 mixture instead of 10-0 (i.e. pure gas)?
We thank the reviewer for the useful comment. Our goal in this work was to differentiate a single analyte gas from a mixture of different analyte gases. For this, we collected responses of individual analytes and their mixtures in different proportions.
- How are the various datasets (train, validation, test) made for the SVM? On line 172 it is written that the SVM is trained with an artificial database. It is important to understand how many data and which ones (experimental or artificial) each dataset contains.
We thank the reviewer for the useful com ment. The train and validation datasets were artificially constructed by the database expansion process. And the test dataset was obtained from an experiment. Those datasets for train, validation, and test contain 720, 180 and 9 data, respectively. To clarify this, we added this information in the revised manuscript.
(Page 10, line 267)
(720 train data, 180 validation data)
- Only one quantitative result is given regarding the classification: on line 173 it is said that the total accuracy is 95.24%. The authors should specify the accuracy for each type of measurement: each individual gas and each of the mixtures, to understand which classes are easier and more difficult to discriminate.
We agree with the reviewer that we should specify the accuracy for all the tests performed. The number of testsets and their identification accuracy under the suggested protocol are charted in the table below. Individual vapors were identified with 100% accuracy, but the one for mixture vapors was 91.67%. Identification for mixture vapors involving HCHO vapor is lower accuracy than others. As suggested, we add this Table S1 in the Supporting Information.
(Page 9, line 244)
The number of testsets and their identification accuracy under the suggested protocol are charted in the Table S1. Individual vapors were identified with 100% accuracy, but the one for mixture vapors was 91.67%. Identification for mixture vapors involving HCHO vapor is lower accuracy than others.
(Table S1 )
Table S1. Identification accuracy of individual vapors and mixture vapors
|
Type |
Vapors |
Test |
Results |
Accuracy |
|
Individual |
NO2 |
9 |
9 |
100% |
|
NH3 |
9 |
9 |
100% |
|
|
HCHO |
9 |
9 |
100% |
|
|
Mixture |
NO2+NH3 |
9 |
9 |
100% |
|
NH3+HCHO |
9 |
8 |
89% |
|
|
HCHO+NO2 |
9 |
8 |
89% |
|
|
NO2+NH3+HCHO |
9 |
8 |
89% |
- The plots in Fig. S3 show strong drift, higher than the response and recovery of the various measures. Why does this not negatively affect the performance of the electronic nose?
We agree with the reviewer that the drift affects the measurement of the response amplitude and the recovery of the sensors. However, the result obtained by applying the suggested protocol showed high recognition accuracy over 90%. And this is because what matters in the measurements is how different the resistances of the sensor elements are, not how accurate their values are.
(Page 9, line 250)
The suggested protocol is also shown to be robust against the drift of the response baseline.
- The SVM classifier is not described in detail (for example: what kind of kernel does it use?)
We used the Linear SVM to identify individual and mixture vapors, which does not require any kind of kernels. Other types of SVM (polynomial, sigmoid and gaussian) were considered, however they typically need a hyper parameter for a tuning purpose when a dataset was changed. We thought the Linear SVM was more robust in our protocol, and the obtained identification accuracy was high (~95.24%) enough for the operation. As mentioned by the reviewer, we added the corresponding sentence in the revised manuscript as follows.
(Page 4, line 151)
Also, we used the linear SVM to identify individual and mixture vapors, which does not require any kind of kernels. Other types of SVM (polynomial, sigmoid and gaussian) were considered, however they typically need a hyper parameter for a tuning purpose when a dataset was changed. We thought the linear SVM was more robust in our protocol, and the obtained identification accuracy was high (~95.24%) enough for the operation.
Minor comments:
- Line 69: CSA is not defined (you have defined ICSA some lines before). It would be better to use homogeneous terminology throughout the article.
We thank the reviewer for the comment. We revised the manuscript accordingly.
- Line 72-73: "forming Au, Pd, and Pt islands in nanoscale via the agglomeration of the Au films." I do not think that the agglomeration of Au can create islands of Pd or Pt.
We thank the reviewer for the useful comment. Sensing layers including four thin films and three metal (Au, Pt, and Pd) as catalyst were deposited by e-beam evaporation with an on-axis mode. After annealing at 500℃ for 2 h, the SnO2, TiO2, WO3, and In2O3 films are crystallized and a 3 nm- thick metal film is agglomerated into nanoparticles (NPs), resulting in thin films decorated with NPs on the surface. As shown in Fig.1 below, after annealing, the diameter of self-agglomerated metal NPs on the surfaces of In2O3 and SnO2 films are about 10 nm while it is about 50 nm on TiO2 and WO3. Such variation in size is attributed to the surface energy difference between metal NPs and metal oxides where smaller NPs have higher surface energies. To clarify this, we added the corresponding sentence in the revised manuscript as follows.
(Page 2, line 91)
After annealing at 500 ℃ for 2 h, the WO3, SnO2, TiO2, and In2O3 films are crystallized and a 3 nm- thick metal film is agglomerated into nanoparticles (NPs), resulting in thin films decorated with NPs on the surface. After annealing, the diameter of self-agglomerated metal NPs on the surfaces of In2O3 and SnO2 films are about 10 nm while it is about 50 nm on TiO2 and WO3. Such variation in size is attributed to the surface energy difference between metal NPs and metal oxides where smaller NPs have higher surface energies.
Figure 1. Field-emission scanning electron microscopy (FE-SEM) images of 16 sensor elements. After thermal processes, the functionalized layer of the metals formed nanoscale metallic islands.
- Line 77: “ICSAs were operating at different temperatures” : are they two different arrays or just one that operated first at one temperature and then at another temperature?
We thank the reviewer for the useful comment. A single sensor array was operated sequentially at different temperatures. A micro-heater is mounted on the back of a printed circuit board (PCB) of ICSA and the power consumption was 250 mW. This heater can control the temperature according to the applied voltage. The operation power of our CSA was about 0.87 mW/mm2 (=250 mW/(13 mm× 22 mm)) at 200 ℃ and 0.77 mW/mm2 (=220 mW/(13 mm× 22 mm)) at 150 ℃. Therefore, we were able to analyze the response of the ICSA for two temperatures (150 and 200℃) with a single sensor. To clarify this, we added the corresponding sentence in the revised manuscript as follows.
(Page 3, line 117)
A micro-heater is mounted on the back of a printed circuit board (PCB) of ICSA and the power consumption was 250 mW. This heater can control the temperature according to the applied voltage. The operation power of our CSA was about 0.87 mW/mm2 (=250 mW/(13 mm× 22 mm)) at 200 ℃ and 0.77 mW/mm2 (=220 mW/(13 mm× 22 mm)) at 150 ℃.
- Line 84: “were aged”: in what conditions were they aged? At normal working condition (150 or 200 ° C?) Or at higher temperature to stabilize the crystal structure?
We thank the reviewer for the comment. The CSA module underwent an aging process at 150 ℃ and 200 ℃ for 72 h to make the sensing materials thermally stable at each measurement temperature. For details, we added the corresponding sentence in the revised manuscript as follows.
(Page 3, line 107)
The CSA module underwent an aging process at 150 ℃ and 200 ℃ for 72 h to make the sensing materials thermally stable at each measurement temperature.
- Line 88: what does “auto-functionalized mass flow controller” mean?
As shown in Fig. 1 below, with the gas flow change from dry air to target gases, the variations of the resistances in our sensor array were simultaneously monitored using a digital sourcemeter (Keithley 2635A). The various concentrations of NH3, HCHO, and NO2 vapors were controlled in high precision by an auto-functionalized mass flow control (MFC). For example, to make 25 ppm NH3 gas using 50 ppm NH3 gas, the flow rate ratio of ammonia gas and air gas should be 1000 sccm: 1000 sccm (1:1). At this time, auto-functionalized mass flow controller means that MFC automatically adjusts the flow rate. Consequently, we added the corresponding sentence in the revised manuscript as follows.
Figure 1. Gas sensing measurement equipment with auto-controlling humidity system.
- Line 89: "With vapor flow change from ambient air to target chemical vapors": but at the beginning of the paragraph you say that you used dry air. Have you used ambient air or dry air?
We appreciate this comment. We used dry air (with a humidity of 5% or less) in the measurements. We corrected the manuscript to clarify this.
(Page 3, line 107)
The fabricated ICSA was examined in dry air (with a humidity of 5% or less) using a chamber with the volume of 12800 cm3 (16 cm (W)×16 cm (H)×50 cm (L)).
- Line 109: “when the response is saturated”: in Fig. 4 it does not seem that saturation is ever reached
We agree that this expression is not clear to understand. Thus, we decided to remove the above sentence “when the response is saturated”.
- After Fig. 1, Fig. 5 is mentioned in the text. Perhaps it would be worth changing the order of the figures so that they go in order with the text.
We thank the reviewer for the comment. We revised the manuscript accordingly.
- Line 112 "The normalized response and its direction give the amplitude and polarity of the input matrices for PCA." What does “direction” mean? You probably mean if the resistance increases or decreases, but since you made the absolute value in the equation on line 108, the Vnormalhas no "direction". And what are the “amplitude and polarity” of matrices for PCA? The matrix for the PCA should simply consist of 9 lines for each gas, each line consisting of the 32 response values obtained from the various sensors.
We appreciate this comment. As mentioned by the reviewer, "direction" means an increase or decrease in the sensor response signal (voltage). As we revised the manuscript, we confirmed that there was an error in the response of the ICSA (VR) equation. Therefore, the equation was replaced as below:
"amplitude and polarity" means the magnitude and increase or decrease of the voltage. We replaced the corresponding sentence with the following
(Page 4, line 135)
The normalized response and the increase or decrease in resistance provide the amplitude and polarity of the input matrices for PCA.
- Line 114: “One hundred data samples were randomly generated…”: one hundred in total or one hundred for each gas?
We appreciate this comment. One hundred data samples were randomly generated for a single analyte vapor at a certain concentration. To clarify this, we revised the manuscript.
(Page 4, line 137)
One hundred data samples were randomly generated for a single analyte vapor at a certain concentration in the range of −2σ to +2σ from the distribution (σ: standard deviation).
- The random error to generate virtual data is applied to each individual sensor individually, and not correlated with the error on the others, right?
We appreciate this comment. Since the sensor elements in the array are electrically disconnected from each other, we do not think there is any possible correlation between their random errors.
- Line 130: "The obtained analytical matrix has 300 by 32 structure for each vapor." -> do you mean for each mixture? What about the 3-gas mixture?
We appreciate this comment. The obtained analytical matrix is for each individual vapor. As answered in question 22, one hundred data sample generated for a single analyte vapor at a certain concentration. Then, including three different concentrations leads to 300 by 32 matrix structure. Since the artificial matrix for 3-gas mixture was constructed by using the matrices of individual vapors. Therefore, it has 300 by 32 matrix structure as well.
- Fig S4: the labels of the lines are missing (I imagine they are the 32 sensors composing the array, but they should be specified to better understand the effect of material, decoration and working temperature on the response.
We agree that Fig. S4 is not clear to understand. Thus, we decided to remove the Fig. S4.
- Principal components are ordered with the% variance they bear. The difference between PC1 and PC2 is very high (92.3 and 0.8% respectively). Can the authors explain this behaviour between the importance of PC1 and all 31 other PCs?
We thank the reviewer for the comment. We think that this indicates one of the sensor elements has exceptionally good response characteristics to differentiate analyte vapors. To improve this, we have to find good sensing materials or catalysts in terms of selectivity.
- Lines 162 - 164: from the 9 experimental points for each gas you have created 91 "virtual" points for each gas. How are they divided into test, training and validation datasets? The points in Fig. 6 are the test ones, I guess.
We thank the reviewer for the comment. As answered in question 9, The train, validation and test dataset were decided by random sampling of the 80:20:1 ratio. The points in Fig. 6 are the test dataset. The structure of matrix for each vapor has 300 by 32. we randomly selected the datasets for training and validation and the test datasets were obtained experimentally.

Reviewer 4 Report
The title of manuscript is “Identification of chemical vapor mixture assisted by artificially extended database for environmental monitoring”. The objective of the study was suggest a novel method to distinguish chemical vapors of individuals and even their mixtures and demonstrate it with a fully integrated sensor array. The suggested protocol is composed of principal component analysis (PCA), artificial database construction, and classification. The Authors demonstrate that this method significantly reduces the required amount of training data using artificially extended database for the identification of chemical vapor mixtures with very high accuracy.
The Introduction of the study provides with some general information about the aroma techniques analysis. On the other hand for several decades, studies on application of different type of techniques for detection of odor have been conducted. I suggest supplementing the Chapter with additional information related to other new methods and devices in research of VOCs detections. „Performance analysis of mau-9 electronic-nose mos sensor array components and ann classification methods for discrimination of herb and fruit essential oils”; “Opto-electronic nose coupled to a silicon micro pre-concentrator device for selective sensing of flavored waters”; “E-nose coupled with an artificial neural network to detection of fraud in pure and industrial fruit juices”.
Additional information contained in the Introduction chapter will make the aim of the study will clearly stated.
The Materials and Methods section provides the reader with enough information to repeat the experiments conducted. The strength of the work is the use of advanced statistical methods to analyze the obtained results. I have a series of questions about the Principal Component Analysis (PCA) used in the work:
On the basis of which criterion was the optimal number of main components obtained in the PCA analysis determined?
How many columns and rows had a data matrix for PCA?
Is the input matrix automatically scaled?
Please put this information in this chapter.
In the Results and discussion chapter contains information should be supplemented on discussing with the other items from the last years of publication including problems of reactions on the chemical compounds.
The conclusions are well and were supported by the data.
The literature used is appropriate but should be supplementing about the items from the last years of publication. References should be provided as required by Sensors.
Author Response
The title of manuscript is “Identification of chemical vapor mixture assisted by artificially extended database for environmental monitoring”. The objective of the study was suggest a novel method to distinguish chemical vapors of individuals and even their mixtures and demonstrate it with a fully integrated sensor array. The suggested protocol is composed of principal component analysis (PCA), artificial database construction, and classification. The Authors demonstrate that this method significantly reduces the required amount of training data using artificially extended database for the identification of chemical vapor mixtures with very high accuracy.
The Introduction of the study provides with some general information about the aroma techniques analysis. On the other hand for several decades, studies on application of different type of techniques for detection of odor have been conducted. I suggest supplementing the Chapter with additional information related to other new methods and devices in research of VOCs detections. „Performance analysis of mau-9 electronic-nose mos sensor array components and ann classification methods for discrimination of herb and fruit essential oils”; “Opto-electronic nose coupled to a silicon micro pre-concentrator device for selective sensing of flavored waters”; “E-nose coupled with an artificial neural network to detection of fraud in pure and industrial fruit juices”.
Additional information contained in the Introduction chapter will make the aim of the study will clearly stated.
The Materials and Methods section provides the reader with enough information to repeat the experiments conducted. The strength of the work is the use of advanced statistical methods to analyze the obtained results. I have a series of questions about the Principal Component Analysis (PCA) used in the work:
On the basis of which criterion was the optimal number of main components obtained in the PCA analysis determined?
We thank the reviewer for the comment. We considered 32 principal components (PCs), and the percentages of the variance of PC1 and PC2 are 92.3% and 0.8%, respectively. Since the variance of other PCs are very low compared to those, we proceeded the protocol with PC1 and PC2 for the identification process.
We determined the number of main components by considering the percentage of variance and identification accuracy. The data matrix has 32 columns in structure, and accordingly, 32 PCs were obtained in PCA. The percentage of variance indicating the loss of information in converting the data into the principal component space is determined by the calculation of eigenvalues. From the calculation, it turned out that PC1 and PC2 determines 93.1% of the response pattern. Therefore, we decided to use PC1 and PC2 for the suggested protocol. When we confirmed the identification accuracy using higher order PCs, the identification accuracy was not improved. As suggested, we added the corresponding sentence in the revised manuscript.
(Page 7, line 195)
We determined the number of main components by considering the percentage of variance and identification accuracy. The data matrix has 32 columns in structure, and accordingly, 32 PCs were obtained in PCA. The percentage of variance indicating the loss of information in converting the data into the principal component space is determined by the calculation of eigenvalues. From the calculation, it turned out that PC1 and PC2 determines 93.1% of the response pattern. Therefore, we decided to use PC1 and PC2 for the suggested protocol. When we confirmed the identification accuracy using higher order PCs, the identification accuracy was not improved.
How many columns and rows had a data matrix for PCA?
Thank you for your comment. The matrix for each analyte vapor at a certain concentration has 100 by 32 elements in structure after the data expansion process. Since we worked with the data at three different concentrations and tried to differentiate 7 cases (3 individual analyte vapors and 4 different combinations of them), this makes it 2100 by 32 matrix structure. Consequently, we added the corresponding sentence in the revised manuscript as follows.
(Page 7, line 202)
The matrix for each analyte vapor at a certain concentration has 100 by 32 elements in structure after the data expansion process. Since we worked with the data at three different concentrations and tried to differentiate 7 cases (3 individual analyte vapors and 4 different combinations of them), this makes it 2100 by 32 matrix structure.
Is the input matrix automatically scaled?
Thank you for your comment. The matrix is not automatically scaled. We performed a normalization step to scale the signal of ICSA. The Min-Max Normalization method was used in this study. The normalization was performed according to the following equation:
In this study, three individual gases and four mixture gases were identified using 32 sensors. To achieve this, the normalization process was done on the dataset including all the responses at different concentrations in different combinations of analyte vapors. Consequently, we added the corresponding sentence in the revised manuscript as follows.
(Page 4, line 141)
In addition, the matrix is not automatically scaled. We performed a normalization step to scale the signal of ICSA. The Min-Max Normalization method was used in this study (Equation (2)). Three individual gases and four mixture gases were identified using 32 sensors. To achieve this, the normalization process was done on the dataset including all the responses at different concentrations in different combinations of analyte vapors.
Please put this information in this chapter.
In the Results and discussion chapter contains information should be supplemented on discussing with the other items from the last years of publication including problems of reactions on the chemical compounds.
The conclusions are well and were supported by the data.
The literature used is appropriate but should be supplementing about the items from the last years of publication. References should be provided as required by Sensors.

Round 2
Reviewer 3 Report
The article, whose substance was already good from the start, was greatly improved by adding the scientific details that were needed.
English still has several errors but it is understandable.
I do not entirely agree with some of the authors' answers, but this does not affect the value of the manuscript in its current form.